# Two-Dimensional Plasmons in Laterally Confined 2D Electron Systems

**DOI:** 10.3390/nano13060975

**Published:** 2023-03-08

**Authors:** Igor V. Zagorodnev, Andrey A. Zabolotnykh, Danil A. Rodionov, Vladimir A. Volkov

**Affiliations:** 1Kotelnikov Institute of Radio-Engineering and Electronics of the RAS, 125009 Moscow, Russia; 2Moscow Institute of Physics and Technology, 141701 Dolgoprudny, Russia

**Keywords:** two-dimensional plasmon, quantum well, Drude model, plasmon polaritons in confined two-dimensional electron systems, confined magnetoplasmon, confinement of gated plasmon, near-gate plasmon

## Abstract

The collective oscillations of charge density (plasmons) in conductive solids are basic excitations that determine the dynamic response of the system. In infinite two-dimensional (2D) electron systems, plasmons have gapless dispersion covering a broad spectral range from subterahertz to infrared, which is promising in light-matter applications. We discuss the state-of-the-art physics of 2D plasmons, especially in confined 2D electron systems in stripe and disk geometry, using the simplest approach for conductivity. When the metal gate is placed in the vicinity of the 2D electron system, an analytical description of the plasmon frequency and damping can be easily obtained. We also analyze gated plasmons in the disk when it was situated at various distances from the gate, and discuss in detail the nontrivial behavior of the damping. We predict that it is not a simple sum of the radiative and collisional dampings, but has a nonmonotonic dependence on the system parameters. For high-mobility 2D systems, this opens the way to achieve the maximal quality factor of plasma resonances. Lastly, we discuss the recently discovered near-gate 2D plasmons propagating along the laterally confined gate, even without applied bias voltage and having gapless dispersion when the gate has the form of a stripe, and discrete spectrum when the gate is in the form of disk. It allows for one to drive the frequency and spatial propagation of such plasmons.

## 1. Introduction

In this article, we review and analyze the properties of the plasma oscillations or plasmons in two-dimensional (2D) electron systems (ESs) in which carriers, electrons or holes, are localized in a very thin layer or plane and are free to move in two spatial dimensions. Though they have already been studied for more than half a century [1], scientific interest is undying due to their exceptional fundamental properties and promising applications. Most recently, they have been intently discussed with regard to graphene and other novel 2D materials. Being a basic collective excitation, they are at the interdisciplinary crossroad of condensed matter physics and electrodynamics. Advances in any of these fields, and the technology and fabrication of nanostructures, allow for creating plasmonic structures with new and versatile properties.

The spectrum of 2D plasmons in infinite 2DESs was first predicted theoretically [1,2]. Then, plasmons were observed experimentally for 2DES on a liquid helium surface [3] and in silicon inversion layers [4,5]. Two-dimensional plasmons possess a gapless dispersion law, in contrast to the ”conventional” plasmons in 3D conducting materials [6], e.g., metals. This resembles the surface plasmon polaritons at a metal–dielectric interface, but the dispersion law of 2D plasmons can be controlled by applying the voltage to a planar metal electrode (gate) placed near 2DES or an external magnetic field. However, surface plasmon polaritons in ultrathin conductive films [7,8] or 2D materials [9] are 2D plasmons. Due to the rather low 2D carrier concentration and the simultaneous high quality of modern 2DESs, 2D plasmon frequencies may lie in giga- and terahertz ranges, which are very attractive for applications. The plasmons can be used for generators and detectors of terahertz radiation [10,11,12,13,14,15,16,17], they limit the operational range of a number of devices [18], and are also interesting for their fundamental properties [19,20,21,22,23,24,25,26,27,28,29,30,31,32,33,34].

To utilize 2D plasmons, it is necessary to clearly understand their properties, especially in profiled nanostructures, which are used ubiquitously for effective light-plasmon interaction. Although the main features of 2D plasmons are generally well-known, in recent years, a number of important fundamental properties have been recognized, especially in practically essential laterally confined 2DESs, which we review. The best-quality plasmons have been obtained in high-mobility quantum wells and heterostructures such as GaAs/AlGaAs structures in which the electron relaxation time due to collisions with impurities and phonons reaches record values [35]. Nevertheless, most of the results are also valid for new conductive 2D materials, e.g., high-quality graphene, in the frequency range where conductivity can be described with the Drude formula [36,37].

We describe plasmons in the simplest classical manner using Maxwell’s equations and the dynamical Drude conductivity for the constitutive relation between the current density and electric field in 2DES. The dynamics of confined 2D plasmons is controlled by partial differential equations, which can be reduced to a complex integrodifferential equation. Of particular interest are the analytical solutions of this integral equation under conditions close to the experimental situations. We pay special attention to the damping of 2D plasmons because it determines the range of existence of the plasmons and the feasibility of their utilization in applications. First, in Section 2, we consider the plasmons in infinite 2DESs with a gate, taking into account electromagnetic retardation and a constant external (nonquantazing) magnetic field directed perpendicular to the 2DES plane. As a first example of a confined structure, in Section 3, we analyze the plasma modes of a stripe in the vicinity of the gate when an exact solution is achievable. After that, in Section 4, we consider plasmons in a disk at a different distance from the gate, and analyze in detail the dependence of plasmon damping on the parameters of the system. We show that, for confined geometries such as disks and stripes, plasma frequencies can be estimated with the phenomenological quantization of the wave vector in the dispersion for the infinite system. However, the radiative damping rate and thereby the overall damping of the plasmon resonances in the finite-size 2DESs cannot be described via this procedure. Then, in Section 5, the fully analytical theory of near-gate plasmons in partially gated 2DES is presented. We end with conclusions in Section 6.

## 2. Infinite System

Before considering the confined structures, it is instructive to discuss an infinite 2DES, because its plasmon spectra can be easily derived analytically, which helps to understand qualitatively many features of plasmons in confined 2DESs. We considered a 2DES situated at a distance *d* above the gate, which was assumed to be an ideal infinite metal plane. The system was placed into a perpendicular constant magnetic field B.

We considered the propagation of 2D plasma excitations in the form of plane waves, i.e., the solutions for (small) charge density and other deviations from their average values being proportional to $\exp\left(-i\omega t+i\mathrm{qr}\right)$, where ω and q are the plasmon frequency and the wave vector lying in the 2DES plane, respectively. In what follows, without loss of generality, we assumed that plasmon frequency had a positive value.

In an isotropic 2DES, the conductivity tensor is antisymmetric, i.e., $\sigma_{xx}=\sigma_{yy}$, $\sigma_{xy}=-\sigma_{yx}$. Then, the plasmon dispersion equation has the following form [38,39,40,41]:(1)$$F\left(\frac{i\omega \sqrt{\varepsilon }}{c\beta }\right)F\left(\frac{c\beta }{i\omega \sqrt{\varepsilon }}\right)+{\left(\frac{2\pi \sigma_{xy}}{c\sqrt{\varepsilon }}\right)}^{2}=0,\;\mathrm{where}\;F\left(\eta \right)=\frac{2\pi \sigma_{xx}}{c\sqrt{\varepsilon }}\eta -\frac{1}{1-e^{-2\beta d}}.$$
where ε is the dielectric permittivity of the surrounding medium (ε is the same between 2DES and the gate, and in the half-space without the gate), *c* is the speed of light in vacuum, and parameter $\beta =\sqrt{q^{2}-\varepsilon {\left(\omega /c\right)}^{2}}$, which possesses a positive real part, Reβ≥0 [42], describes the localization of electromagnetic fields near the 2DES in the upper half-space. We use the CGS system of units throughout the paper.

Let us consider several well-known cases. If the external magnetic field is absent, then $\sigma_{xy}=0$, and Equation (Equation 1) splits into two independent equations describing the transversal and longitudinal modes. The transverse-electric (TE) mode, which is determined with equation $F(i\omega \sqrt{\varepsilon }/c\beta )=0$, exists only in systems with $Im(\sigma_{xx})<0$. For example, it takes place in narrow frequency range near the interband transitions frequencies in graphene [43,44] and other Dirac materials [45,46]. The solution of the second equation, $F(c\beta /i\omega \sqrt{\varepsilon })=0$, corresponds to the transverse-magnetic (TM) mode. It exists for $Im(\sigma_{xx})>0$; consequently, at all those frequencies in which there is no TE solution, it is, in fact, a very broad frequency range. Therefore, in this article, we only deal with these ”usual” 2D TM-plasmons.

Further we use the isotropic dynamical Drude model:(2)$$\sigma_{xx}=\frac{\left(\gamma -i\omega \right)}{{\left(\gamma -i\omega \right)}^{2}+\omega_{c}^{2}}\frac{ne^{2}}{m},\;\sigma_{xy}=\frac{-\omega_{c}}{{\left(\gamma -i\omega \right)}^{2}+\omega_{c}^{2}}\frac{ne^{2}}{m},$$
where *n*, −e, and *m* are the 2D electron concentration, charge, and effective mass, respectively, γ=1/τ is the collisional decay rate, τ is the electron relaxation time due to collisions with impurities and phonons, and $\omega_{c}=eB/mc$ is the cyclotron frequency. The Drude approximation is suitable if the plasmon frequency is much less than the Fermi energy (divided by *ħ*) and the interband transition frequencies, and the wavelength of the plasmon is much greater than the Fermi wavelength and the electron mean free path. In practice, this approximation works well for a typical 2DES, based on quantum well or graphene structures, at least up to terahertz frequencies [19,47,48,49].

For further discussion, we first consider the so-called case of ”ungated” 2DES where the distance between the 2DES and gate *d* tends to infinity. For simplicity, we neglected the electromagnetic retardation effects, i.e., we focused on the quasielectrostatic (or quasistatic) approximation in which the plasmon frequency is much smaller than the light frequency at the same wave vector. In the absence of an external magnetic field and assuming that the collisional decay rate γ is small compared to the frequency of plasmons, i.e, taking into account corrections only of the first order in γ, we obtain from Equations (Equation 1) and (Equation 2) the following dispersion of the ungated 2D plasmons:(3)$$\omega \approx \omega_{p}\left(q\right)-i\frac{\gamma }{2},\;\mathrm{where}\;\;\;\omega_{p}\left(q\right)=\sqrt{\frac{2\pi ne^{2}q}{m\varepsilon }}.$$

Here, the imaginary part of the frequency describes the damping of the plasmon. To obtain high-quality plasma resonance, it must be much less than the real part of the frequency. In the considered basic case, the damping is described by the electron relaxation time, which is an intrinsic characteristic of a particular 2DES; therefore, there is a limitation on the plasma frequencies available for observation in the 2DES. Thus, 2D plasma resonances in gigahertz range can be detected in high-quality 2DESs such as quantum wells at low (helium) temperatures; at terahertz and higher frequencies, these resonances can already be observed in high-quality graphene structures [22,50,51]; in ultrathin metal films, the resonances appear only at frequencies close to those of visible light [8].

When the 2DES is placed into a perpendicular constant magnetic field, 2D plasmons are usually referred to as magnetoplasmons. In the first order in γ and in the quasistatic approximation, their dispersion for the ungated 2DES has the following form:(4)$$\omega \left(B\right)\approx \sqrt{\omega_{p}^{2}\left(q\right)+\omega_{c}^{2}}-i\frac{\gamma }{2}\left(1+\frac{\omega_{c}^{2}}{\omega_{c}^{2}+\omega_{p}^{2}\left(q\right)}\right),$$

Thus, the finite value of the magnetic field opens the frequency gap in the spectrum of 2D magnetoplasmon. At high magnetic fields, the plasmon frequency asymptotically approaches the cyclotron resonance. However, beyond the quasistatic limit, when one take into account the finite value of the speed of light (i.e., the electromagnetic retardation effects), the magnetodispersion crosses the cyclotron line $\omega =\omega_{c}$: as the magnetic field increases, the frequency tends asymptotically to a finite value independent of the magnetic field.

The damping of the ungated 2D magnetoplasmons, i.e., the imaginary part of the frequency (Equation 4), increases continuously from γ/2 to γ as the magnetic field increases and it is always proportional to collisional decay rate γ. In a homogeneous infinite 2DES, plasmons do not emit electromagnetic waves; therefore, it has only ohmic loses described with the electron relaxation time. From a formal standpoint, when deriving Equation (Equation 1), we considered only the nonradiative evanescent waves. The edges of confined structures are inhomogeneities and necessarily lead to plasmon radiation, which we discuss in the following sections.

Now, we analyze the other important case in which the distance to the gate *d* is much less than the plasmon wavelength. This is called the fully screened limit because the long-range Coulomb interaction between electrons in 2DES is suppressed, and electrons effectively interact locally with their images in the metal gate. In this case, we expanded the exponential term $\exp\left(-\beta d\right)$ in function F(η) in (Equation 1) to the first order via parameter βd and found that the TM plasmon had linear dependence on the wave vector [52,53]:(5)$$\omega \approx \frac{v_{p}}{\sqrt{1+\varepsilon \frac{v_{p}^{2}}{c^{2}}}}q-i\frac{\gamma }{2\left(1+\varepsilon \frac{v_{p}^{2}}{c^{2}}\right)},\;v_{p}=\sqrt{\frac{4\pi ne^{2}d}{\varepsilon m}},\;qd\ll \sqrt{1+\varepsilon \frac{v_{p}^{2}}{c^{2}}}$$

When deriving this equation, we assumed that the damping was small enough and left only the first-order terms in the decay rate. The resulting ”gated” 2D plasmons have constant velocity. Quasistatic speed $v_{p}$ is usually less than the speed of light, but it is not necessary [53]. The plasmon damping is proportional to the collisional decay rate, but also depends on speed $v_{p}$ and decreases with an increase in $v_{p}$. The reason is that, in quasistatic limit $v_{p}\ll c$, the plasmon energy is stored only in the form of the kinetic energy of the charges and their potential interaction. When parameter $v_{p}$ increases, the energy is also stored in electromagnetic fields surrounding the 2DES plane. The plasmon frequency approaches the light frequencies (at the same wave vector), i.e., the plasmon transforms into a plasmon–polariton, and at the same time, the damping decreases, since the plasmon electromagnetic field is not subjected to ohmic losses in the nondissipative dielectric environment under consideration.

The fully gated 2D magnetoplasmon dispersion has the following form in the long-wavelength limit βd≪1 [41,54]:(6)$$\omega \left(B\right)\approx \sqrt{\frac{\omega_{c}^{2}}{{\left(1+\varepsilon \frac{v_{p}^{2}}{c^{2}}\right)}^{2}}+\frac{v_{p}^{2}q^{2}}{1+\varepsilon \frac{v_{p}^{2}}{c^{2}}}}-i\frac{\gamma }{2\left(1+\varepsilon \frac{v_{p}^{2}}{c^{2}}\right)}\left(1+\frac{\omega_{c}^{2}}{\omega_{c}^{2}+v_{p}^{2}q^{2}\left(1+\varepsilon \frac{v_{p}^{2}}{c^{2}}\right)}\right).$$

This expression is applicable at frequencies much lower than the frequency of light (at the same wave vector). At high magnetic fields, the plasmon frequency reaches a constant independent of the magnetic field [41]. The magnetoplasmon frequency is again related to the plasmon frequency without a magnetic field (Equation 5) and the cyclotron frequency, but the cyclotron frequency is renormalized with parameter $v_{p}/c$. This renormalization manifests itself, for example, in the absorption spectra of normally incident radiation [55]. The damping of the fully gated 2D magnetoplasmon is somewhat less then γ/2 and changes by a factor of 2 with an increasing magnetic field according to Equation (Equation 6).

Now, we examine plasmons in confined structures.

## 3. Fully Screened Stripe

In this section, we consider plasmons in a 2DES in the form of a stripe surrounded by a dielectric medium with permittivity ε. Let the stripe have infinite length and finite width *W*. The infinite metal gate is placed under the stripe at distance *d*. For 2DES conductivity, we used the Drude model (Equation 2). Let us direct the *x* and *y* axes of the Cartesian coordinate system across and along the stripe, respectively, with the origin at the center of the stripe. The longitudinal component of a wave vector along the *y* axis, $q_{\left|\right|}$, is conserved. The electromagnetic fields, the current, and the perturbation of the charge density oscillate as $exp(iq_{\left|\right|}y-i\omega t)$.

Using Maxwell’s equations, the local Ohm’s law, and the vanishing current at the edges of the stripe, we obtained the integrodifferential equation for the current density [56]:(7)$$j\left(x\right)=\frac{i}{\varepsilon \omega }\left(\begin{array}{ccc} \sigma_{xx} & \; & \sigma_{xy}\\ -\sigma_{xy} & \; & \sigma_{xx} \end{array}\right)\left(\begin{array}{cc} \frac{\partial^{2}}{\partial x^{2}}+\varepsilon \frac{\omega^{2}}{c^{2}} & iq_{\left|\right|}\frac{\partial }{\partial x}\\ iq_{\left|\right|}\frac{\partial }{\partial x} & -q_{\left|\right|}^{2}+\varepsilon \frac{\omega^{2}}{c^{2}} \end{array}\right)\int_{-\frac{W}{2}}^{\frac{W}{2}}dx^{\prime }G(|x-x^{\prime }\left|\right)j\left(x^{\prime }\right),$$
where the integral kernel is
(8)G(ξ)=∫−∞∞dqxeiqxξβ1−e−2dβ,β=qx2+q||2−εω2c2,Reβ≥0.

Let us consider the fully screened limit. For that, we expand Equation (Equation 8) up to the first order in distance *d*, which turns the integral kernel $G(|x-x^{\prime }|)$ into delta function $4\pi d\delta (x-x^{\prime })$. This significantly simplifies the analysis because Equation (Equation 7) becomes purely differential and can easily be solved exactly with zero-current boundary conditions. As a result, we find plasma frequencies that are characterized by a non-negative integer *N* corresponding to the number of half-wavelengths that fit in the stripe width. The dispersion of the plasma modes with N≠0 is given exactly by Equation (Equation 6), where wave vector *q* is equal to $\sqrt{q_{\left|\right|}^{2}+{\left(N\pi /W\right)}^{2}}$. The lowest (fundamental) mode N=0 has the following dispersion:(9)$$\omega_{0}=\frac{v_{p}\left|q_{\left|\right|}\right|}{\sqrt{1+\varepsilon \frac{v_{p}^{2}}{c^{2}}}}-i\frac{\gamma }{2\left(1+\varepsilon \frac{v_{p}^{2}}{c^{2}}\right)}.$$

This does not depend on the magnetic field at all. In the considered approximations, the current of this mode flowed strictly along the stripe. In fact, this mode is the ”fully screened” edge magnetoplasmon (oscillating in the same phase at both edges of the stripe). In the next order, in distance *d*, it would have negative magnetodispersion [57].

The plasmons are nonradiative in this case (and their dampings are proportional to γ) since the radiation of oscillating electrons in the stripe is fully suppressed by the radiation of their images in the metal.

## 4. Gated Disk

Consider a conductive disk of radius *R* in the plane z=0 at a distance *d* above the metal gate. The system is schematically presented as an inset in Figure 1. Given the cylindrical symmetry, the plasmon modes can be characterized with angular momentum l=0,±1,±2,… and radial number $n_{r}=1,2,\ldots$. These numbers characterize the ”quantization” of the plasmon wave vector along the perimeter and the radius of the disk, respectively. In the absence of a magnetic field, the mode with l=0 corresponds to the axisymmetric oscillations where charges and currents move only in the radial direction. Sometimes, these modes are also called dark or breathing modes, since they have zero dipole and nonzero electric quadrupole moments, and thereby interact rather weakly with electromagnetic radiation [58,59]. The mode with the lowest frequency (the fundamental mode) has l=1 and $n_{r}=1$. The current of this mode flows through the center and along the edge of the disk. For this reason, to approximate the frequency of the fundamental mode and other modes with l≠0 and $n_{r}=1$, the wave vector should rather be quantized by the perimeter of the disk, i.e., q∼l/R, while for the axisymmetric mode, it is quantized by the diameter of the disk, i.e., q∼π/R. This also explains why the fundamental, and not the axisymmetric, mode is l=1.

After these preliminary remarks, we first turn to the simplest fully screened case.

### 4.1. Fully Screened Limit

Here, we examine the disk plasmons in the limit of d→0, when the plasmon dispersion equation is transcendent and is given by the following expression [53,60]:(10)$$\xi \frac{d}{d\xi }\ln\left(J_{l}\left(\xi \right)\right)+\frac{l\omega_{c}}{i\gamma +\omega \left(1+\varepsilon \frac{v_{p}^{2}}{c^{2}}\right)}=0,$$
where
(11)$$\xi^{2}=\frac{\omega R^{2}}{v_{p}^{2}}\left[i\gamma +\omega \left(1+\varepsilon \frac{v_{p}^{2}}{c^{2}}\right)-\frac{\omega_{c}^{2}}{i\gamma +\omega \left(1+\varepsilon \frac{v_{p}^{2}}{c^{2}}\right)}\right]$$
and $J_{l}\left(\xi \right)$ is the Bessel function of the first kind of the *l*-th order.

Without a magnetic field, the frequency of the plasmon eigenmodes is
(12)ωl,nr(0)=vp1+εvp2c2μl,nrR−iγ21+εvp2c2,Reωl,nr≫γ,
where $\mu_{l,n_{r}}$ is the $n_{r}$-th zero of the first derivative of the *l*-th order Bessel function. Comparing it with the plasmon dispersion in infinite system (Equation 5), we see that, in the fully screened limit, the appropriate quantization of the wave vector in the disk is $\mu_{l,n_{r}}/R$.

In the weak magnetic field, the frequency is given by
(13)ωl,nrωc=ωl,nr(0)1+lωcRμl,nrμl,nr2−l2vp1+εvp2c2,Reωl,nr(ωc)≫γ.

Dispersion Equation (Equation 10) is invariant only under simultaneous signs changing l→−l and $\omega_{c}\to -\omega_{c}$. In the presence of a magnetic field, it leads to the splitting of the dispersion with the sign of the orbital momentum *l* [61]. Qualitatively, this can be understood as follows. The sign of the angular momentum actually corresponds to the direction of the charge rotation: clockwise or counterclockwise. The magnetic field also tends to rotate the electrons in a certain direction. Thus, depending on whether these rotations are codirected or not, the magnetic field either increases the plasma resonance frequency or decreases. With a further increase in the magnetic field, the frequency of modes with $n_{r}>1$ (and l<0) begins to increase, while the magnetodispersion of modes with $n_{r}=1$ (and l<0) remains negative, i.e., the permanent decrease occurs only for modes with l<0, $n_{r}=1$. In fact, these modes are edge magnetoplasmons [57,62,63,64,65]. In high magnetic fields, the perturbation of their charge density is mainly localized near the disk edge, and their frequency can be written as follows:(14)$$\omega_{l<0,n_{r}=1}\left(\omega_{c}\to \infty \right)\to \frac{v_{p}}{\sqrt{1+\varepsilon \frac{v_{p}^{2}}{c^{2}}}}\frac{\left|l\right|}{R}-i\frac{\gamma }{2\left(1+\varepsilon \frac{v_{p}^{2}}{c^{2}}\right)}.$$

Since the edge magnetoplasmon currents in a strong magnetic field are localized in the vicinity of the disk edge, their wave vector is quantized only by the perimeter of the disk. The magnetodispersion of the other ”bulk” modes tends towards the renormalized cyclotron resonance:(15)$$\omega_{l,n_{r}}\left(\omega_{c}\to \infty \right)\to \frac{\omega_{c}}{1+\varepsilon \frac{v_{p}^{2}}{c^{2}}}-i\frac{\gamma }{1+\varepsilon \frac{v_{p}^{2}}{c^{2}}}.$$

This tendency was recently seen in an experiment [66]. As we can see from Equation (Equation 15), in high magnetic fields and in the considered fully screened limit, the damping of the edge magnetoplasmon is twice less than that of the bulk plasmons.

For the purpose of illustration, Figure 1 shows the magnetodispersion of the plasma frequencies (Equation 10) in quasistatic limit $\left(\varepsilon v_{p}^{2}\approx 0.02c^{2}\right)$. The figure also displays the order of the plasmon modes in the absence of a magnetic field, namely, l=±1→±2→0 as the frequency increases.

### 4.2. Finite Distance to Gate

Now, let us consider arbitrary distances *d* between the disk and the gate without an external magnetic field. In the previous sections, we discussed the plasma oscillations as eigenmodes of a system. Here, we use another approach—the excitation of plasma resonances with external electromagnetic radiation. In the absorption spectra, the maxima indicate the excited plasma resonances, while their linewidths are related to the damping of the plasma modes. However, the full width at half maximum we used is twice the damping as the imaginary part of the eigenmode frequency.

Before moving on, let us clarify what parameters determine the plasma resonances in the disk. The 2DES in the form of a disk without a gate is governed by three independent ”intrinsic” parameters: disk radius *R*, collisional damping rate γ, and quantity $ne^{2}/m$ in the conductivity (the Drude weight). The frequency of the external radiation, ω, is the extraneous parameter. With appropriate nondimensionalization (all frequencies multiplied by $\sqrt{\varepsilon }R/c$), the plasma resonances are determined by only two independent dimensionless parameters: dimensionless damping rate $\gamma \sqrt{\varepsilon }R/c$, and the so-called retardation parameter [60]:(16)$$\tilde{\Gamma }=\frac{2\pi ne^{2}R}{mc^{2}}=\frac{1}{2}\varepsilon \frac{v_{p}^{2}}{c^{2}}\frac{R}{d},$$
because the denominator of this expression contains the speed of light. Indeed, applying Dispersion Equation (Equation 1) with phenomenological quantization rule q∼1/R, one can ensure that, when this parameter is small, then the quasistatic frequency is obtained; if $\tilde{\Gamma }\gg 1$, then a plasmon–polariton with frequency ω∼c/R is obtained. We explicitly show this behavior in the next section. Lastly, this nondimensionalization can also be applied when there is a metal gate near the disk if one use dimensionless parameter d/R.

Now, let us focus on the main axisymmetric resonance that can be excited by an oscillating point dipole over the center of the disk with the dipole moment directed perpendicular to the 2DES plane. The dependence of axisymmetric resonance frequency $\omega_{0,1}$ and its linewidth $\Delta \omega_{0,1}$ on the gate distance *d* are shown in Figure 2. When the metal is moved from the disk, the frequency monotonously increases in accordance with Equation (Equation 12) and approaches the asymptotic value corresponding to the ungated limit, which we discuss in the following section. At the same time, at small distance *d*, when the local interaction limit is applicable, the linewidth shows a reduction, which is in accordance with Equation (Equation 12). Then, it increases and exhibits decaying oscillations, reaching the asymptotic value of the ungated disk.

The occurrence of the linewidth oscillation can be easily explained. The presence of the gate near the disk affects the radiation losses due to the interference of the electromagnetic radiation emitted by the disk and the metal. The latter can be regarded as the fields created by the ”image disk”—the mirror reflection of the actual disk of the metal plane that has the opposite currents and charges. Thus, it results in constructive or destructive interference depending on the distance between the disk and the gate, which manifests itself in the oscillation of the linewidth.

It is instructive to consider the linewidth in more detail in the limit of the small separation distance, d≪R. Using an approximate solution method, we found the following [67]:(17)$$\Delta \omega_{0,1}\approx \gamma \left(1-2\tilde{\Gamma }\frac{d}{R}+4{\tilde{\Gamma }}^{2}\frac{d^{2}}{R^{2}}+\ldots \right)+156{\left(\frac{d}{R}\right)}^{5}{\tilde{\Gamma }}^{4}\frac{R}{\sqrt{\varepsilon }c}+\ldots \;.$$

The ”collisional” part proportional to γ exactly matches the corresponding expansion of the imaginary part in Equation (Equation 12) at small *d*. The only difference is in the last term of Equation (Equation 17) that accounted for the plasmon radiation. In our case, this term is the octupole radiation of the system. The axisymmetric mode in the ungated limit has a zero dipole moment ($\propto {\tilde{\Gamma }}^{2}$) and a nonzero quadrupole moment ($\propto {\tilde{\Gamma }}^{3}$) [60]. However, the nearby gate suppresses the quadrupole radiation, rendering the octupole radiation ($\propto {\tilde{\Gamma }}^{4}$) dominant. Such a radiative contribution cannot be described with the dispersion law of any infinite system at all since, in its derivation, only nonradiative (localized near 2DES) modes are considered. This is a unique feature of confined structures.

### 4.3. Ungated Limit

Let us now examine the frequency and the linewidth of the disk plasma resonances at various values of the retardation parameter in the ungated limit, i.e., in the absence of a metal gate. In Figure 3, we plotted the dependence of the frequency and the linewidth of the main axisymmetric resonance. The frequency monotonically depends on the retardation parameter; at small values of the parameter, $\tilde{\Gamma }\ll 1$, it has a square-root dependence on $\tilde{\Gamma }$ in agreement with Equation (Equation 3). At large retardation, $\tilde{\Gamma }\gg 1$, the frequency approaches a constant value, $const\times c/\sqrt{\varepsilon }R$, which corresponds to the plasmon–polariton regime. The linewidth has less trivial behavior. In ”clean” limit γ=0, the plasmon damping consists only of the radiative losses and increases monotonically with increasing retardation parameter (the lower blue curve). Because the axisymmetric mode is quadrupole, it behaves as ${\tilde{\Gamma }}^{3}$ at the beginning. When the collision decay rate is nonzero, the dependence is nonmonotonic with the minimum at some point (the upper blue curve).

Let us discuss the appearance of the minimum in the linewidth dependence in detail. The damping and thereby linewidth can be estimated as the ratio of the power loss to the energy stored in the plasma oscillations:(18)$$\Delta \omega =\frac{P_{J}+P_{rad}}{E_{kin}+E_{em}},$$
where $P_{J}$ and $P_{rad}$ are the Joule and radiative power losses, respectively, $E_{kin}$ is the kinetic energy of the plasma oscillations (electrons) in the disk, and $E_{em}$ is the electromagnetic energy of the plasma oscillations. We estimate all these quantities in the weak retardation regime. For the axisymmetric plasma mode, we have only radial current component $j_{r}\left(r\right)$. Then, the Joule loss is equal to
(19)PJ=12∫0RRe|jr(r)|2σxx2πrdr=2π2γRc2Γ˜∫0R|jr(r)|2rdr.

Due to the symmetry of charge and current distributions, the electric and magnetic dipole moments of the axisymmetric mode are zero. Therefore, at the small retardation, the quadrupole moment *Q* gives the main contribution to the radiative power. In the Cartesian coordinate system, it is $Q=\frac{2\pi }{i\omega }diag\left(1,1,-2\right)\int_{0}^{R}j_{r}\left(r\right)r^{2}dr$, where diag(×) is a diagonal matrix. Then, the radiative power is
(20)$$P_{rad}\approx \frac{\omega^{6}{\left(\sqrt{\varepsilon }\right)}^{3}}{180c^{5}}\frac{{\left|Q\right|}^{2}}{2}=\frac{\pi^{2}{\left(\sqrt{\varepsilon }\right)}^{3}\omega^{4}}{15c^{5}}|\int_{0}^{R}j_{r}\left(r\right)r^{2}dr{|}^{2},$$
where ${\left|Q\right|}^{2}$ is a trace of the product of the matrix *Q* and its complex conjugate.

Now, let us calculate the stored energy. Since we considered the weak retardation regime, we started with the quasistatic limit and found corrections to it. For convenience, we considered the time moment when the current is maximal. In that case, the kinetic energy of electrons in the disk can be written as follows:(21)$$E_{kin}=\int_{0}^{R}n\frac{m}{2}{\left(\frac{|j_{r}\left(r\right)|}{ne}\right)}^{2}2\pi rdr=\frac{2\pi^{2}R}{c^{2}\tilde{\Gamma }}\int_{0}^{R}{\left|j_{r}\left(r\right)\right|}^{2}rdr.$$

In quasistatics, at the time when the current is maximal, the perturbation of the charge density is equal to zero, and there are no electromagnetic fields around the disk. In the next order via retardation, the magnetic field appears, and the electromagnetic energy is stored in it. We used the connection of the magnetic field with the current density
(22)$$H_{\theta }\left(r,z\right)=\frac{2\pi }{c}\frac{\partial }{\partial z}\int_{0}^{R}\left(\int_{0}^{\infty }\frac{pdp}{\beta }e^{-\beta |z|}J_{1}\left(pr\right)J_{1}\left(pr^{\prime }\right)\right)j_{r}\left(r^{\prime }\right)r^{\prime }dr^{\prime },\;H_{r}\left(r,z\right)=H_{z}\left(r,z\right)=0,$$
where Imβ<0, since this corresponds to the outgoing electromagnetic radiation from the disk. Taking into consideration that $\sqrt{\varepsilon }\omega R/c\ll 1$ in the quasistatic limit, the electromagnetic energy can be approximated as follows:(23)$$E_{em}\approx \frac{1}{4\pi }\int_{-\infty }^{\infty }\int_{0}^{\infty }{\left|H_{\theta }\left(r\right)\right|}^{2}2\pi rdrdz=\frac{\pi^{2}}{c^{2}}\int_{0}^{R}\int_{0}^{R}G\left(r,r^{\prime }\right)j_{r}\left(r\right)j_{r}^{*}\left(r^{\prime }\right)rr^{\prime }drdr^{\prime },$$
where $G\left(r,r^{\prime }\right)=\int_{0}^{\infty }J_{1}\left(pr\right)J_{1}\left(pr^{\prime }\right)dp$.

We expressed each quantity in Equation (Equation 18) in terms of the current; now, we need an appropriate expression for it. The simplest approximation $j_{r}\left(r\right)=r/R\left(1-r^{2}/R^{2}\right)$ is extremely suitable [60]. Calculating all integrals and taking into account that, in the quasistatic limit $\omega_{0,1}R/c\approx 1.87\sqrt{\tilde{\Gamma }/\varepsilon }$, we arrived at the following final expression:(24)$$\Delta \omega_{0,1}\approx \frac{\gamma }{1+0.17\tilde{\Gamma }}+\frac{0.068{\tilde{\Gamma }}^{3}}{1+0.17\tilde{\Gamma }}\frac{c}{\sqrt{\varepsilon }R}.$$
We find that the collisional term $P_{J}/\left(E_{kin}+E_{em}\right)$ decays while the radiative contribution $P_{rad}/\left(E_{kin}+E_{em}\right)$ increases with the increase of the retardation parameter. Thus, the above mentioned nonmonotonic behavior is connected with the competition between the collisional and the radiative losses. The obtained expression (Equation 24) can help to understand qualitatively the drastic reduction of the linewidth recently discovered in the experiment [68] in the strong retardation regime.

The linewidth of the plasma resonance is not just a sum of the collisional damping $\tilde{\gamma }$ rate and the radiative contribution ($\propto {\tilde{\Gamma }}^{3}$ for the quadrupole axisymmetric mode and $\propto {\tilde{\Gamma }}^{2}$ for the dipole fundamental mode). There is an additional term depending on both the collision decay rate and the retardation parameter. Its presence leads to the local minimum of the linewidth dependence on the retardation parameter.

## 5. Partially Gated Infinite 2D Electron Systems

This section is devoted to an interesting case that is an ”intermediate” between the cases of gated and ungated 2DES. Namely, we consider plasmons in the so-called partially gated 2DES when 2DES is infinite in both directions, while the gate, which represents the thin metal layer situated in the vicinity of 2DES at a small distance *d*, has a finite size in one or both directions. Specifically, we focus on the cases of the gate in the form of a stripe with infinite length and a disk. In this section, unlike the previous ones, we consider plasma waves in frames of quasielectrostatic approximation without taking into account retardation effects, and we also assume that 2DES is ”clean”, i.e., the electron relaxation time due to collisions with impurities and phonons is infinitely large. There was no applied gate voltage, so 2DES is homogeneous with constant electron concentration *n*.

In such partially gated 2DESs, a new type of plasma excitations appears, the so-called near-gate or proximity plasmons. This new type of plasmons was first discovered theoretically [69] and experimentally [70,71,72] for the gate in the form of a stripe, so let us consider this geometry in more detail; see Figure 4a. In this case, the plasmon frequency is governed by two ”quantum numbers”: the first is the continuous wave vector along the stripe $q_{\left|\right|}$, and the second, *N*, takes on integer non-negative values 0,1,2,… and corresponds to the number of zeros of the plasmon charge density across the stripe. In [69], under the conditions of d≪W, $q_{\left|\right|}d\ll 1$, and under the assumption that the frequency of sought-for plasmons was small compared to the frequency of ungated 2D plasmons with wave vector $q_{\left|\right|}$, the following dispersion equation was derived:(25)$$k{\left(\tan\frac{kW}{2}\right)}^{\pm 1}=\pm \left|q_{\left|\right|}\right|,$$
where *W* is the width of the stripe, $k=\sqrt{\omega^{2}/v_{p}^{2}-q_{\left|\right|}^{2}}$, $v_{p}$ is the quasistatic velocity of gated plasmons (Equation 5), and upper and lower signs define the dispersion equation for even and odd modes with respect to the charge density across the stripe. The fundamental near-gate plasmon mode that had no nodes of the charge density in the across-the-stripe direction, i.e., N=0, in the long wavelength limit $q_{\left|\right|}W\ll 1$ had a square-root dispersion law that was unexpected for gated systems [69]:(26)$$\omega =\sqrt{\frac{4\pi ne^{2}d}{m\varepsilon }\frac{2|q_{\left|\right|}|}{W}},$$
where ε is the dielectric permittivity of the media. This simple analytically obtained expression for plasmon frequency was in a very good agreement with the experimental data [70,71]. Let us emphasize the ”hybrid” form dispersion in (Equation 26). On the one hand, $\omega \propto \sqrt{q_{\left|\right|}}$ is for ungated plasmons (Equation 3), and on the other, $\omega \propto \sqrt{d}$ corresponds to the case of gated plasmons (Equation 5). The analytically found spectrum of near-gate plasmons from Equation (Equation 25) is shown in Figure 4c. Along with the fundamental mode N=0, there are higher modes with N=1,2,… that have finite frequencies at zero wave vector along the stripe. These modes at small $q_{\left|\right|}$ lay inside the continuum of usual ungated 2D plasmons existing far from the gate, and the border of the continuum is defined by relation $\omega =\omega_{p}\left(q_{\left|\right|}\right)$; see Equation (Equation 3). Higher near-gate plasmon modes, therefore, interact with the continuum of 2D plasmons, resulting in the finite lifetimes of these modes due to decay into the continuum. Such modes are often referred to as quasistationary. However, if one neglects the above mentioned decay, the asymptotic behavior of the frequency in the long wavelength limit, $q_{\left|\right|}W\ll 1$, is as follows:(27)$$\omega_{N}^{2}=v_{p}^{2}\left(\frac{\pi^{2}N^{2}}{W^{2}}+\frac{4|q_{\left|\right|}|}{W}\right),\;N=1,2,\ldots$$

Modes that lay outside the continuum of 2D plasmons are localized near the gate in the across direction. In contrast to higher modes, fundamental mode N=0 is always localized in the long wavelength limit, as its frequency (Equation 26) is below $\omega_{p}$: $\omega /\omega_{p}\left(q_{\left|\right|}\right)=\sqrt{4d/W}\ll 1$. An interesting feature of near-gate plasmons is that their charge density is almost entirely situated under the gate, while the electric potential and electric field component laying in the 2DES plane extend beyond the gated region to distances on the order of $1/q_{\left|\right|}$. This qualitatively explains the inbetween nature of these plasmons compared to that of gated and ungated ones. The fundamental mode can be described in the frame of the lumped element approach; see the supplemental material of [70]. In short wavelength limit $q_{\left|\right|}W\gg 1$, all modes have almost linear asymptotic behavior described by $\omega_{N}^{2}=v_{p}^{2}\left(\pi^{2}{\left(N+1\right)}^{2}/W^{2}\right)+v_{p}^{2}q_{\left|\right|}^{2}$. Lastly, if one takes into account the finite value of the gate conductivity, instead of square-root dispersion (Equation 26), the fundamental mode in the long wavelength limit possesses a linear spectrum with the characteristic velocity depending on the value of gate conductivity [73].

Now let us move to the case of plasmons in 2DES with the disk-shaped gate [74,75,76,77]; see Figure 4b. In this case, the frequencies of near-gate plasmons are governed by discrete numbers; the first is orbital number *l*, and the second is radial number *N*. For disk-shaped gates, all plasmon modes demonstrate quasistationary behavior [75], i.e., they have finite lifetimes due to decay into the continuum of ungated 2D plasmons. If the lifetimes are large compared to the inverse frequencies of the modes, and one neglects the decay, then the plasmon frequencies can be described with the following dispersion equations [74,75]:(28)$$\bar{\omega }J_{l}^{\prime }\left(\bar{\omega }\right)+\left|l\right|J_{l}\left(\bar{\omega }\right)=0\;\mathrm{at}\;l\neq 0$$
and
(29)$$\bar{\omega }\ln\left(\frac{e^{C}{\bar{\omega }}^{2}d}{R}\right)J_{0}^{\prime }\left(\bar{\omega }\right)-J_{0}\left(\bar{\omega }\right)=0\;\mathrm{at}\;l=0,$$
where $J_{l}$ is the Bessel function of the first kind and *l*th order, $\bar{\omega }=\omega R/v_{p}$, and *C* is the Euler–Mascheroni constant. For the l=0 mode, the bottom surface of the gate charges in the process of oscillations. In the case of an electrically isolated gate, as in Figure 4b, this means that the charges are redistributed between the top and bottom surfaces of the gate, while the effect of charges from the upper surface on the 2D electrons is neglected. In the case of structures in which a disk-shaped gate is electrically connected to 2DES with a wire, the excitation of the mode with l=0 is essential since it determines most of the low-frequency electromagnetic responses of such structures [78,79]. The plasmon frequencies defined by Equation (Equation 28) were in good agreement with those obtained experimentally [76].

The fact that near-gate plasmons in 2DES with a disk-shaped gate are quasistationary indicates that they should reveal themselves as peculiarities in the scattering of usual 2D plasmons on the gated region of 2DES. Indeed, the scattering cross-section have a series of pronounced resonances with frequencies corresponding to the frequencies of the near-gate plasmons defined with Equations (Equation 28) and (Equation 29) [75]. The angular dependence of the differential cross-section allows to determine the orbital number of the corresponding near-gate plasmon mode, while the width of the resonance line defines the inverse lifetime of the mode. Besides 2DESs with a disk-shaped gate, the scattering can be used to characterize the frequencies and lifetimes of quasistationary near-gate plasmons in 2DESs with various gate shapes, including stripes and more complicated forms.

If a partially gated 2DES with an ideal metal gate is placed in the weak perpendicular magnetic field B, the near-gate plasmon frequency at zero magnetic field ω(B=0) is modified as usual for the quasistatic plasmons (Equation 4):(30)$$\omega \left(B\right)=\sqrt{\omega^{2}\left(B=0\right)+\omega_{c}^{2}}.$$

Above, we did not take into account collisional decay rate γ. The collisions lead to the collisional damping of near-gate plasmons. In the absence of a magnetic field, in the quasistatic regime, and assuming that the frequency of near-gate plasmons is much greater than that of γ, this damping rate equals γ/2 in the case of ungated 2D plasmons as well (Equation 3).

Lastly, let us emphasize once again that, in the considered partially gated 2DESs, there was no bias voltage applied to the gate, so the conductivity of the 2DES was homogeneous. The existence and (quasi)localization of near-gate plasmons were entirely due to the different Coulomb interactions between 2D electrons inside and outside the gated area of the 2DES.

## 6. Conclusions

In summary, we considered and reviewed the recent theoretical studies of plasma waves in confined 2DESs on the basis of quantum wells and heterostructures. We assumed that an ideal metal electrode (gate) could be situated parallel to the 2DES, and/or a constant external magnetic field could be applied perpendicular to 2DES plane. In this paper, we focused on plasmons in 2DES in the form of a stripe or a disk, or an infinite 2DES with laterally restricted gate.

Concerning 2DESs in the form of a stripe or a disk, we found that electromagnetic retardation effects significantly modify the plasmon frequency and the lifetime. The latter was contributed by the so-called collisional damping appearing due to electron scattering on impurities and phonons, and the radiative damping that occurred due to the radiation of electromagnetic waves during plasmon oscillations. Although the frequency of plasma waves could be qualitatively estimated from the plasmon dispersion law for an infinite 2DES, the damping required much more accurate analysis. In particular, the damping is not a simple sum of collisional and radiative dampings; instead, it has a nonmonotonic dependence on system parameters. For plasmons in gated 2DESs, the damping can have prominent part oscillating as a function of the distance between the 2DES and the gate, which can significantly affect the plasmon lifetime even if this distance is large compared to the lateral size of 2DES and the plasmon wave length.

We also briefly discussed the recently discovered so-called near-gate 2D plasmons that appear in infinite homogeneous 2DES with a laterally restricted gate situated in its vicinity. For the stripe-shaped gate, the fundamental mode of such plasmons possesses gapless dispersion low (the dependence of the frequency on the wave vector along the gate) and shows unexpected properties, combining the features of gated and ungated plasmons. In the case of a disk-shaped gate, near-gate plasmons had a discrete spectrum and manifested a strong interaction with the ”bulk” of ungated 2D plasmons, which resulted in the finite lifetimes of such modes due to the decay in the continuum of the 2D plasmons existing far from the gate.

## Figures and Tables

**Figure 1 nanomaterials-13-00975-f001:**
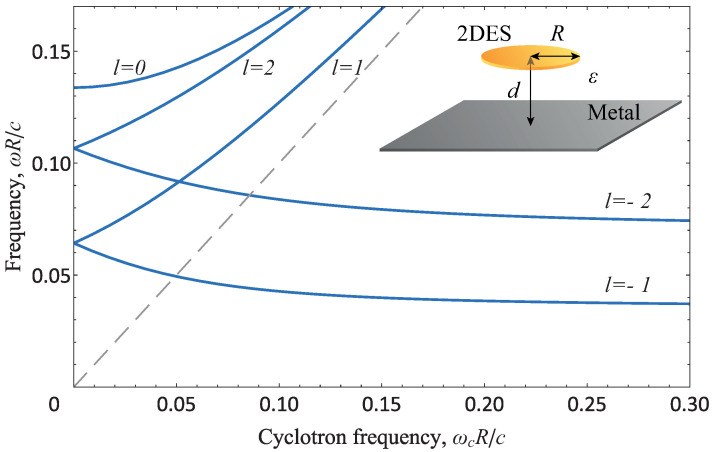
Dependence of the plasmon frequency ω of the disk with radius R=100 μm at the distance d=5 μm to the metal gate on the dimensionless cyclotron frequency $\omega_{c}=eB/mc$ for the lowest modes with l=0,±1,±2 and $n_{r}=1$. The data were obtained for the collisionless limit (γ→0) at a typical concentration for GaAs/AlGaAs quantum wells $n=6\times 10^{11}\;{\mathrm{cm}}^{-2}$, dielectric constant ε=12.8, and effective mass $m=0.067\;m_{0}$, where $m_{0}$ is the free electron mass. The dashed straight line visualizes the cyclotron resonance to which the upper modes tend in strong magnetic fields. The inset illustrates the system under consideration.

**Figure 2 nanomaterials-13-00975-f002:**
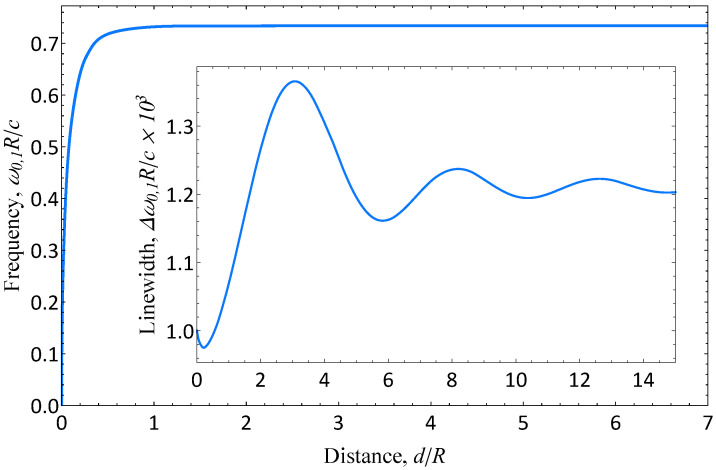
Dependence of main axisymmetric resonance frequency $\omega_{0,1}$ and its linewidth $\Delta \omega_{0,1}$ on the distance *d* between the disk and the gate for retardation parameter $\tilde{\Gamma }=0.16$, collisional decay rate γR/c=0.001, and dielectric constant ε=1.

**Figure 3 nanomaterials-13-00975-f003:**
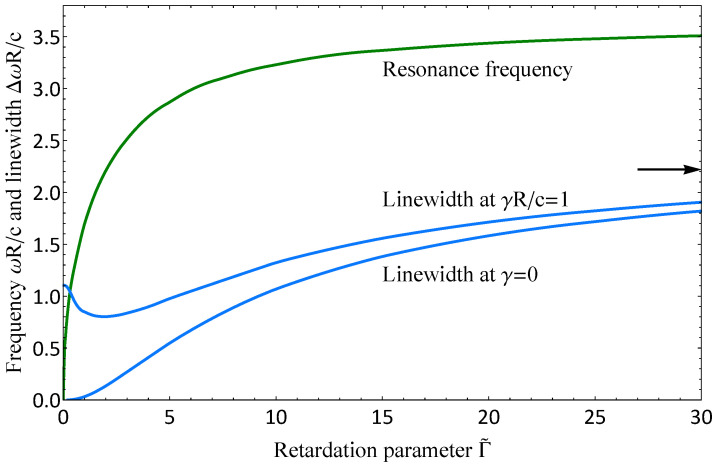
Dependence of the frequency $\omega_{0,1}$ (green) and linewidth $\Delta \omega_{0,1}$ (blue) of the main axisymmetric plasma resonance in an ungated disk placed in a vacuum (ε=1) on the retardation parameter. Since the resonance frequency slightly depends on the retardation parameter, it is shown at γ=0. The linewidth is presented at γ=0 (the bottom blue curve) and γR/c=1 (the upper blue curve). The arrow indicates the asymptotic value of the linewidth at $\tilde{\Gamma }\to \infty$.

**Figure 4 nanomaterials-13-00975-f004:**
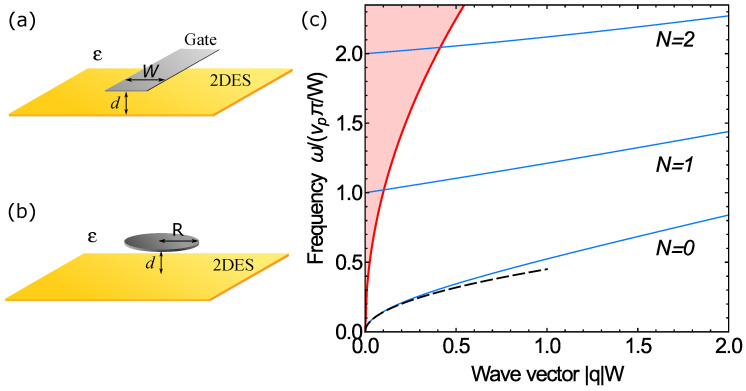
Schematic views of 2DES with a gate in the form of (**a**) a stripe and (**b**) a disk. The distance between the gate and 2DES is denoted by *d*, while *W* and *R* are the width of the stripe and the radius of the disk, respectively. We assumed that R≫d and W≫d. 2DESs and gates were embedded into the media with constant dielectric permittivity ε. (**c**) Blue lines represent the analytically found spectrum of near-gate plasmons for 2DES with the stripe-shaped gate; see Equation (Equation 25). The red line denotes the boundary of the 2D plasmon continuum $\omega =\omega_{p}\left(q_{\left|\right|}\right)$ (see Equation (Equation 3)), within which the higher modes N=1,2,… have finite lifetimes. The dashed line shows the long wavelength asymptote (Equation 26) of the fundamental mode N=0. Relation d/W have a value of 0.005.

## Data Availability

Data sharing not applicable.

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
