# Peer review of "Two-Dimensional Plasmons in Laterally Confined 2D Electron Systems"

_nanomaterials, 2023, doi:10.3390/nano13060975_

Round 1

Reviewer 1 Report

In the manuscript “2D plasmons in laterally confined 2D electron systems” by I. V. Zagorodnev et al. is  a short review paper on the properties of plasma oscillations in two-dimensional (2D) electron systems confined in a thin layer. In particular, the authors describe the influence on the in-plane collective excitations of the lateral confinement, magnetic field, and the near-by metallic gate of different shapes. The authors pay special attention to the plasmon damping since this important characteristic determine the range of existence, propagation length of the plasmon charge oscillations, as well as feasibility of the plasmon utilization in practical applications. The main topics discussed in the manuscript are mainly based on the recent results obtained essentially by the same authors with collaboration with an experimental research group. In essence, the modifications in the 2D plasmon dispersion and damping caused by externally applied magnetic field, the gate, and the shape of both 2D system and the gate are described analytically. In particular, the quantization of the plasmonic mode in the laterally confined systems is thoroughly described.

From my point of view, the manuscript is clearly written and well structured. The results are presented in a convincing manner and the main relevant references are appropriately cited. The conclusions are supported by the results discussed in the manuscript. Therefore, I suggest publishing this manuscript.

I only suggest correcting several points:

1. Definition of dielectric function of the dielectric media permittivity \eps should be after a first appearance in Eq. (1) instead of at Line 164.

2. It is not clear how around Eq. (7) the momentum q_|| pointing along the stripe is defined in respect to the x and y coordinates. Also, I guess that the momentum q defined in the section “Partially gated infinite 2D electron systems” is different from q employed in Section 2 for the infinite 2D system. It looks it is more close to q_||.

Author Response

We cordially thank the reviewer for the useful questions and comments.

Please, find our answers below.

  1. Definition of dielectric function of the dielectric media permittivity \eps should be after a first appearance in Eq. (1) instead of at Line 164.

Though there was the definition of the dielectric permittivity before Eq. (1) in the beginning of the Section 2, we moved it immediately after Eq. (1). We hope that the new version of our manuscript is clearer.

  1. It is not clear how around Eq. (7) the momentum q_|| pointing along the stripe is defined in respect to the x and y coordinates. Also, I guess that the momentum q defined in the section “Partially gated infinite 2D electron systems” is different from q employed in Section 2 for the infinite 2D system. It looks it is more close to q_||.

In the revised version of the manuscript we clarify the direction of the momentum q|| in respect to the axes x and y before Eq. (7) in lines 158-160. The wave vector is directed along the y-axis.

We agree with the reviewer that in the section “Partially gated infinite 2D electron systems” the plasmon wave vector along the stripe-shaped gate is much better to notate as q|| instead of q. In the new version of the manuscript we substitute q by q|| in this section.

Reviewer 2 Report

The authors present a review of plasmon oscillation in confined 2D system. They discuss the properties of disk and stripe shaped 2DES and gated systems.

The paper is interesting for researchers working in the field of plasmonics and 2DES, and I suggest the publication after minor revisions

·        To what practical materials the results shown apply? Can the author add experimental results corroborating the theoretical derivation for the plasmon frequencies?

·        The authors cite (line 50-52) that “most of the results are also valid for ….2D materials …graphene”. Can the author comment on that?

·        In figure 4 the authors show the frequency and the wave vector, both in terms of W. What W refer to (total energy of plasma oscillation)? and why do they use such normalization?

·        In figure 1,2 and 3 the authors present the plasmon frequency in terms of R (radius) and c (light speed). Although this is a general representation for 2DES of arbitrary dimension, it does not show in which region of the electromagnetic spectrum the plasmon frequency lies. Can the authors comment on that?

Author Response

We sincerely thank the reviewer for the useful questions and comments.

Please, find our answers below.

To what practical materials the results shown apply? Can the author add experimental results corroborating the theoretical derivation for the plasmon frequencies?

The presented theoretical results should be applied, first of all, to plasmons in 2D systems based on high-mobility semiconductor structures, for example, AlGaAs/GaAs quantum wells. The thing is, the conductivity of these structures is usually well described by the dynamical Drude model (see Ref. [48] of the manuscript), which is crucial for our study. 

Moreover, in semiconductor quantum wells and heterostructures the electron relaxation time can reach 40 ps (at helium temperature), allowing one to observe plasmons in the wide frequency range from several THz down to several GHz.

On the whole, the plasmon frequencies obtained analytically by the use of the presented approach (i.e. the solution of Maxwell’s equations or Poisson's equation + the use of the Drude model for conductivity of 2D system) are in good agreement with experimental observations. Plasmons in 2DES with metal grating Refs. [4,5], edge plasmons and magnetoplasmons Refs. [62-64], gated plasmons in the regime of strong electromagnetic retardation [53], near-gate plasmons Refs. [70,76] are very well described by this simple analytical approach.

In the new version of the manuscript, we emphasize that the experimental data corroborate the derived plasmon frequencies and add new Refs. [61,66] about experimental studies of magnetoplasmons in a fully-screened disk (please, see the second sentence after Eq. (13) and the first sentence after Eq. (15)), and Ref. [68] about a drastic reduction of the plasmon damping of an ungated disk (the last sentence of the paragraph after Eq.(24)). Please also see the sentence after Eq. (26) and the last sentence in the paragraph after Eqs. (28) and (29) in the revised manuscript.

– The authors cite (line 50-52) that “most of the results are also valid for ….2D materials …graphene”. Can the author comment on that?

We meant that our theory works well for systems with conductivity described by the Drude model. For example, in the case of graphene, the Drude model is qualitatively applicable if the frequency is much less than the double Fermi energy divided by Planck's constant.

In the new version of the manuscript we added the phrase and Refs. to clarify this point, please, see the last sentence at the end of the 3rd paragraph in the 1st section (lines 52-54). 

– In figure 4 the authors show the frequency and the wave vector, both in terms of W. What W refer to (total energy of plasma oscillation)? and why do they use such normalization?

In caption to figure 4, the letter W corresponds to the width of the stripe-shaped gate (and so, for example, the product of q_|| and W is dimensionless). Here W is not related to the notations W_{kin} and W_{em} that were introduced in the previous section for the kinetic and the electromagnetic energies of the plasma oscillations, respectively. In the revised version of the manuscript we designate the energies E_{kin} and E_{em} to get rid of this confusion.

– In figure 1,2 and 3 the authors present the plasmon frequency in terms of R (radius) and c (light speed). Although this is a general representation for 2DES of arbitrary dimension, it does not show in which region of the electromagnetic spectrum the plasmon frequency lies. Can the authors comment on that?

The characteristic frequencies of plasmons in 2D semiconductor structures lie in the range of several GHz up to several THz and depend on the electron concentration, the lateral size of the sample (which roughly speaking defines the wavelength), the dielectric permittivity of the surrounding medium, the effective mass of the carriers, the value of the magnetic field, etc.

To give the qualitative estimation of plasmon frequency, consider AlGaAs/GaAs quantum well with parameters as in figure 1, namely, electron concentration equals 6*10^{11} cm^{-2}, dielectric permittivity \epsilon of GaAs is 12.8, effective mass is 0.066m_0, where m_0 is the mass of a free electron. The system has the form of a disk with the radius 100 μm and the distance between the system and the gate is 5 μm. For these parameters we find using Eq. (5) that the quasistatic plasmon velocity v_p equals 1.1*10^9 cm/s, which is much less than the speed of light (c/\sqrt{\epsilon}=8.4*10^9 cm/s). The frequency of the plasmon in the absence of a magnetic field is about 17 GHz.